# The role of health education on cervical cancer screening uptake at selected health centers in Addis Ababa

**Selamawit Hirpa Abu** [1]*, **Berhan Tassew Woldehanna**[1], **Etsehiwot Tilahun Nida**[1], **Abigiya Wondimagegnehu Tilahun**[1], **Mahlet Yigeremu Gebremariam**[2], **Mitike Molla Sisay**[1]

1 School of Public Health, College of Health Sciences, Addis Ababa University, Addis Ababa, Ethiopia,
2 School of Medicine, College of Health Sciences, Addis Ababa University, Addis Ababa, Ethiopia

* Selamawit.hirpa@gmail.com

## Abstract

### Introduction

Cervical cancer is one of the most common causes of morbidity and mortality among women in developing countries including Ethiopia. Unlike other types of cancers, the grave outcomes of cervical cancer could be prevented if detected at its early stage. However, in Ethiopia, awareness about the disease and the availability of screening and treatment services is limited. This study aims to determine the role of health education on cervical cancer screening uptake in selected health facilities in Addis Ababa.

### Methods

Two-pronged clustered randomized controlled trial was conducted in 2018 at eight public health centers that provide cervical cancer screening services using visual inspection with acetic acid (VIA) in Addis Ababa, Ethiopia. Each of the eight health centers were randomly assigned to serve as either an intervention or a control center. A two-pronged clustered randomized controlled trial was conducted in eight public health care centers. All the selected facilities provided cervical cancer screening services using visual inspection with acetic acid (VIA). Four health centers were randomly assigned to the intervention and control arms. The study participants were women aged 30–49 years who sought care at maternal and child health clinics but who had never been screened for cervical cancer. In the intervention health centers, all eligible women received one-to-one health education and educational brochures about cervical cancer and cervical cancer screening. In the control health centers, participants received standard care. Baseline data were collected at recruitment and follow-up data were collected two months after the baseline. For the follow-up data collection, participants (both in the intervention and control arms) were interviewed over the phone to check whether they were screened for cervical cancer.

### Result

From the 2,140 women who participated in the study, 215 (10%) screened for cervical cancer, where 152(71%) were from the intervention health centers. Seventy-four percent of

**Data Availability Statement:** The data are all contained within the manuscript. If any details are needed, I can provide all the data without restriction.

**Funding:** We are grateful for Women's Health Research Working Group at Addis Ababa University, College of Health Sciences for initiating and giving support for this research project. Also, we are very thankful for 'Impact of Maternal Death on Living children' research project at Addis Ababa University, School of Public Health for funding this research. The funders had no role in study design, data collection, and analysis, decision to publish, or preparation of the manuscript.

**Competing interests:** NO authors have competing interest.

these participants reported that they learned about the benefits of screening from the one-to-one health education or the brochure. Women from the intervention health centers had higher odds of getting screened (AOR = 2.43,95%CI;1.58–2.90) than the controls. Women with the educational status of the first degree and those who have a history of sexually transmitted infections (STIs) had higher odds of getting screened (AOR = 2.03,95%CI;(1.15–2.58) and (AOR = 1.55,95%CI;1.01–2.36), respectively.

## Conclusion and recommendation

Providing focused health education supported by printed educational materials increased the uptake of cervical cancer screening services. Integrating one-to-one health education and providing a take-home educational material into the existing maternal and child health services can help increase cervical cancer screening uptake.

## Introduction

Cervical cancer is one of the most common types of cancers among women caused mainly by Human papillomavirus (HPV). In 2018 alone there were an estimated 570,000 new cervical cancer cases globally where 6,294 of the cases occurred in Ethiopia [1, 2]. In countries where there are no routine cervical cancer screening services, women seek health care after they develop the disease manifestations [3]. More than 80% of the cases in sub-Saharan Africa are detected at late stage decreasing the chance of survival [4]. In Ethiopia, an estimated 3,235 cervical cancer related deaths occur annually [4].

Several screening methods are used to detect precancerous lesions and cancer of the cervix. These screening methods are available, cheap, and can be performed safely at outpatient settings [5, 6]. Visual inspection with acetic acid (VIA) screening method is one of the simplest methods applied using household vinegar to see if there is a precancerous lesion. This screening method follows a single visit approach, decreasing treatment seeking related cost and the risk of loss to follow-up. The opportunity this test provides to treat the lesions at the spot is an added benefit as coupling screening with on spot treatment services increases the effectiveness of control programs [7].

Screening programs that have treatment for a positive diagnosis had proved to be a success [8]. Hence, the implementation of VIA is a promising intervention that will help to control the disease. The expansion of the pre-cancer screening and treatment of the lesion using Cryotherapy in Ethiopia affirms the above fact [3]. However, the uptake of cervical cancer screening is very low (19.8%) and absence of disease symptoms was often mentioned for not seeking screening services early on [9, 10].

Community-based health educational programs increase knowledge about cervical cancer and improve cervical cancer screening uptake [10, 11]. Women who have prior knowledge about cervical screening tend to seek cervical screening services compared to those who had no prior knowledge [10]. The Ethiopia Ministry of Health (MOH) recommends educating women who never had cervical cancer screening as a strategy to increase uptake. Using print media is one of the few approaches suggested to reach this target population [3].

In our study, brief face-to-face health education, and brochure which was designed using the Health Belief Model (HBM) was given for the intervention group. Health Belief Model is a theoretical model that can be used to guide health promotion and disease prevention programs. The HBM postulates that people will take action to prevent illness if they consider

themselves as susceptible to a condition (perceived susceptibility), if they believe it would have potentially serious consequences (perceived severity), if they believe that a particular course of action available to them would reduce the susceptibility or severity (perceived benefits), and if they perceive a few negative attributes related to the health action (perceived barriers) [12]. In line with this, the educational brochure and the brief health education discusses what cervical cancer is, who are susceptible to the disease, how to detect the early stage of cervical cancer, how to prevent it and where to find cervical cancer screening services and that they can get the service for free. Thus, this study determined the role of health education on cervical cancer screening uptake at public health care facilities in Addis Ababa.

## Methods

### Study area and period

The study was conducted between August 2017 and January 2018, in Addis Ababa, the capital city of Ethiopia. The 3,147,000 residents of the city live in ten sub-cities and 116 districts [13]. Ethiopia has a three-tier health system where one primary hospital, health centers with its five satellite health posts occupy the lowest level (the primary health care unit). General and referral hospitals constitute the secondary and tertiary levels [14]. This study was conducted in eight health centers which provide cervical cancer screening service using VIA. The health centers are staffed by general practitioners, nurses, and midwives and serve about 250,000 people [14]. The selected health centers were Bole 17, Felegeselam, Lideta, and Addis Ketema primary health centers. The control sites were Kolfe, Kolfe Wordea 09, Arada, and Nifas-Silk health centers.

**Study design and population.** A two-pronged a cluster-randomized controlled trial was conducted among women aged 30–49 years; which is the eligible age group for cervical cancer screening in Ethiopia [3]. This study was conducted in an urban area where most people can read and write or have someone around to read for them. All study participants were selected from Addis Ababa who visited government health centers for family planning (seeking contraceptives or advice for infertility), immunization service, or seeking care for their sick children.

All study participants had no history of cervical cancer screening and were never diagnosed with cervical cancer. Pregnant women and women who gave birth in the past 45 days from the date of data collection were excluded from the study.

### Sample size calculations

The sample size was powered to detect 15% and 10% women who would be screened for cervical cancer at the intervention and control health centers, respectively. By taking the level of significance at 5%, power 90% and 20% loss to follow up; 2,203 study participants were recruited from eight clusters with 230 study participants from each.

**Sampling selection procedure.** At the time of the study, 14 health centers were providing cervical cancer screening in Addis Ababa, all using VIA. Eight high test load health centers were selected based on the number of clients in the past year. The eight health centers were randomized into intervention and control arms, four health centers each.

**Intervention.** Staff nurses working in immunization, family planning, and children clinics were trained by the study team. The trained nurses provided one-to-one health education to all eligible mothers who come seeking care from the clinics within the health centers.

In the intervention health centers, eligible women were recruited from the selected three units (family planning, immunization service, and clinics for infants and children). After mothers were recruited into the study, background information was collected using a structured interview questionnaire. A one-to-one brief health talk lasting 5–10 minutes was provided by health care providers. Definition of cervical cancer, risk factors, susceptibility, mode

of transmission, treatment, and benefits of cervical cancer screening was included in the discussion. When leaving the clinic, a brochure containing information under the topics similar to the brief health talk was issued to the participants. Participants who could not read and write attended the brief health talk but were told to have someone close to them read the content at their convenience. Two months from the time of intervention, participants in the intervention and control groups were communicated over the phone to check whether they were screened for cervical cancer. For those individuals that we could not contact for a follow-up interview, we have tried to reach them through their phone for three consecutive days. If a participant reported that she was not screened, reasons for not screening were asked and those who were screened were asked for the basis of their motivation.

Women in the control group were interviewed using the same standard questionnaire. In addition, only received standard care (did not receive either the one-to-one brief health talk or the educational brochure). According to the session schedule, health education on cervical cancer was supposed to be given twice a week. Early arriving patients and health service clients were the target audiences as health education sessions were held every morning before health care workers commence seeing patients.

**Ethical consideration.**   Ethical clearance was obtained from the Addis Ababa University College of Health Sciences Institutional Review Board (IRB). Permission was obtained from all study health facilities. Each respondent was informed about the purpose and scope of the study. Verbal informed consent as it was approved by the Institutional Review Board; was obtained from recruited study participants prior to their enrolment. The data collectors put a mark on the checklist given on the consent form, regarding the study participants' responses to participate.

**Data entry and analysis.**   Data was entered using Epidata version 3.1 software and later exported to STATA 14 for analysis. The descriptive analysis involved calculating the frequency and percentage of sociodemographic and health service-related variables. Generalized estimating equation (GEE) analysis with a binary response variable using a robust estimator and exchangeable working correlation matrix was used. This was carried out to consider the cluster effect of data among those who received services within the same facility and measure the independent effect of the intervention on the uptake of cervical cancer screening service.

## Result

A total of 2,400 women who had never been screened for cervical cancer prior to this study were recruited. In the follow-up data collection, two thousand one hundred forty participants (89.2%) have participated. Most participants aged between 30 and 40 years with the mean age being 33 (SD±3.8). Eight hundred ninety-eight (85%) of the participants in the intervention group and 973(92%) of the controls were married. Nearly half of the participants in both groups were housewives. Nearly all participants were tested for HIV at least once in their lifetime, where 49(4.9%) from the intervention and 56(5.5%) from the control group were tested positive for HIV. Among those who were in the intervention group and screened for cervical cancer, 73% reported that their source of information was the one to one health education. However, in the control group, 92% of cervical cancer screening related information was obtained from health care providers (Table 1).

In the study period, 215(10%) women were screened for cervical cancer and most were 152 (70.7%) were from the intervention health centers. One to one health education and educational brochure was associated with cervical cancer screening (Table 2).

The frequently mentioned reasons mentioned by participants in the intervention group for not getting screened were lack of time (78%), not having been sick (12%) and not knowing

**Table 1. Socio demographic and health related information of study participants at Addis Ababa, 2017.**

| Variable | | Intervention N = 1,062 (%) | Control N = 1,078 (%) |
|---|---|---|---|
| Age | 30–34 years | 667(63.22) | 753(71.65) |
| | 35–39 years | 280(26.54) | 247(23.50) |
| | 40–44 years | 69(6.54) | 39(3.71) |
| | 45–49 years | 39 (3.70) | 12 (1.14) |
| | Unknown | 0 | 1(0.10) |
| Marital status | Married | 898(85.12) | 973(92.58) |
| | Never married | 67(6.35) | 48(4.57) |
| | Divorced/ Separated | 75(7.11) | 22(2.09) |
| | Widowed | 15(1.42) | 8(0.76) |
| Occupation | Housewife | 553(52.42) | 673(63.97) |
| | Government | 190(18.01) | 118(11.22) |
| | Private | 270(25.59) | 237(22.53) |
| | Other | 42(3.98) | 24(2.28) |
| Woman educational status | Illiterate | 147 (13.93) | 154(14.64) |
| | Primary school | 351(33.27) | 384(36.50) |
| | Secondary school | 356(33.74) | 349(33.17) |
| | Technical and vocational training | 119(11.28) | 99(9.41) |
| | First Degree and above | 82(7.77) | 66(6.27) |
| History of STI | No | 925(88.52) | 957(90.97) |
| | Yes | 120(11.48) | 95(9.03) |
| | Unknown | 10(0.95) | 0 |
| Tested for HIV | No | 50(4.74) | 28(2.66) |
| | Yes | 1,005(95.26) | 1,024(97.34) |
| HIV status | Negative | 956(95.12) | 964(94.14) |
| | Positive | 49(4.88) | 56(5.47) |
| | Unknown | 0 | 4 (0.39) |
| Religion | Ethiopian Christian Orthodox | 742(70.33) | 721(68.54) |
| | Catholic | 6(0.57) | 8(0.76) |
| | Protestant | 103(9.76) | 93(8.84) |
| | Muslim | 203(19.24) | 228(21.67) |
| | Other | 1(0.09) | 2(0.19) |
| Source of information for screening | Radio/Television | 7(4.6) | 1(1.59) |
| | Relative /Friend | 3(3.95 | 6(4.76) |
| | Health Professional | 27(18.42) | 59(92.06) |
| | One to one health education and Brochure | 111(73.03) | 1(1.59) |

**Table 2. Chi- square test result of cervical cancer screening practice and the intervention.**

| Variable | | Screened for cervical cancer | | x2 value | P- value |
|---|---|---|---|---|---|
| | | Yes | No | | |
| | | (n = 215) | (n = 1925) | | |
| | Intervention | 152(70.7%) | 910 (47.3%) | | |
| Type of the health center | | | | 42.45 | 0.000 |
| | Control | 63(29. 3%) | 1,015(52.7%) | | |

about cervical cancer (7%). In the control group lack of time (53%), hasn't been sick (22%) and not knowing about cervical cancer (21%) were mentioned (Table 3).

## Multivariable analysis of the uptake of cervical cancer screening

Variables (Age, marital status, occupation, woman's educational status, history of sexually transmitted infections (STI), HIV testing, HIV test result, religion, allocation to control and intervention group) that have p-value<0.2 with the uptake of cervical cancer screening on bivariate analysis was entered into the multivariable model.

Finally, age, marital status, woman's educational status, occupation history of STI, HIV testing, allocation to control, and intervention group were included in the multivariable analysis. Factors which has a significant association with cervical cancer screening uptake were woman's educational status, history of STI as well as allocation to the control and intervention group. Women from intervention health centers had higher odds AOR = 2.43 (95%CI;1.58–2.90) of getting screened than women from the control health centers. The odds of women with first degree and above to test for cervical cancer is higher AOR = 2.03,95%CI;(1.15–2.58) than the illiterates. Likewise, women who had a history of STI had higher odds to get tested AOR = 1.55,95%CI;(1.01–2.40) (Table 4).

## Discussion

In this study, we found that a one-to-one health talk and issuing brochures as a reminder for women who visited selected health centers increased cervical cancer screening uptake. This was demonstrated among women in the intervention health centers who got the above-mentioned services than women who got only regular health education services in the control health centers.

In developing countries, the main reasons for the low uptake of cervical cancer screening services are attributed to a lack of knowledge about the disease and service availability [15]. Sociocultural factors are also pivotal in determining service uptake [15]. Similarly, in Ethiopia, lack of proper health information from health care providers and low awareness about the disease determined low uptake [9, 10]. In our study integrating one to one health education with maternal and child health care services increased uptake. This approach will help to reach a considerable number of mothers in a short while at an opportune time when they need the information the most. Due to the ongoing antenatal care service-based HIV screening, more than 95% of study participants, in both groups were tested for HIV. Similar to the HIV program if cervical cancer screening is integrated with the routine maternal health service programs like family planning; the uptake would have increased.

**Table 3. Bivariable analysis result for reasons for not being screened for cervical cancer among women in the intervention and control health centers of Addis Ababa.**

| Reasons for not being screened | Health centre type | | P-value |
|---|---|---|---|
| | Intervention | Control | |
| Was busy | 690(78%) | 537(53.6) | 0.000 |
| Never been sick | 107(12%) | 219(21.9) | |
| Don't know the place | 16(1.8%) | 27(2.7) | |
| Don't know about screening | 67(7.5%) | 208(20.8) | |
| Don't like the diagnostic method | 8(0.9%) | 9(0.99) | |
| Service was not available | 2(0.23) | 2(0.2) | |
| **Total** | **890 (100%)** | **1,002 (100%)** | **1,892 (100%)** |

**Table 4. Bivariate and multivariable analysis of intervention effect on uptake of cervical cancer screening.**

| Variable | | Crude OR (95%CI) | Adjusted OR (95%CI) |
|---|---|---|---|
| **Group** | Control | 1 | 1 |
| | Intervention | 2.58(1.66–4.02) | 2.43(1.58–3.9) |
| **Age** | 30–34 years | 1 | 1 |
| | 35–39 years | 0.98(0.70–1.38) | 0.97(0.68–1.37) |
| | 40–44 years | 1.40(0.96–2.89) | 1.5(0.85–2.64) |
| | 45–49 years | 1.89(0.89–4.02) | 1.56(0.73–3.33) |
| **Marital status** | Married | 1 | 1 |
| | Never married | 1.63(0.95–2.79) | 1.45(0.84–2.52) |
| | Divorced/ Separated | 1.57(0.87–2.84) | 1.39(0.77–2.51) |
| | Widowed | 2.15(0.74–6.21 | 2.09(0.71–6.17) |
| **Educational status** | Illiterate | 1 | 1 |
| | Primary school | 0.76(0.48–1.19) | 0.75(0.47–1.20) |
| | Secondary school | 0.97(0.62–1.51) | 0.97(0.61–1.52) |
| | Technical and vocational training | 0.99(0.56–1.74) | 0.94(0.52–1.68) |
| | First degree and above | 2.0(1.15–3.49) | 2.03(1.15–3.58) |
| **Woman occupation** | House wife | 1 | 1 |
| | Government | 2.14(1.48–3.11) | 1.60(1.03–2.49) |
| | Private | 1.14(0.80–1.63 | 1.04(0.71–1.52) |
| | | 1.37(0.63–3.0) | 1.11(0.49–2.49) |
| **Religion** | Orthodox | 1 | - |
| | Catholic | 0.83(0.12–5.55) | |
| | Protestant | 1.47(0.94–2.28) | |
| | Muslim | 1.07(0.74–1.55) | |
| **History of STI** | No | 1 | 1 |
| | Yes | 1.57(1.04–2.36) | 1.55(1.01–2.36) |
| | Unknown | 1.59(0.27–9.37) | 1.27(0.25–6.53) |
| **Tested for HIV** | No | 1 | 1 |
| | Yes | 0.53(0.29–0.97) | 0.57(0.31–1.04) |
| **HIV test result** | Negative | 1 | - |
| | Positive | 1.11(0.59–2.08) | |

The combined one to one health talk and reminders (brochures) contributed to the increase in screening uptake. Another study from Bangladesh indicated the contribution of print media and audiovisuals in improving the awareness and increasing uptake of cervical cancer screening services [16]. Similarly, a systematic review of 66 interventional studies concluded that small media and one-to-one education are effective interventions to increase the uptake of screening for three cancers including cervical cancer [17]. Furthermore, the implementation guideline which was developed through reviewing systematic review studies and expert panel discussion in Ontario, Canada recommended that provision of one-to-one education and assessment of providers are effective interventions for the enhancement of cervical cancer screening uptake at the community-level [18].

In this study, women who were screened for cervical cancer reported that one-to-one health education and the educational brochure that they have received had helped them to decide in screening for cervical cancer. Similar studies have also reported that engaging the community through culturally appropriate health education and reading materials effected a 73% increase in cervical cancer screening uptake [11]. In our study, the intervention doubled the number of people screened for cervical cancer, yet the overall rate remained low. The reason for a smaller

number of women screened in the control groups could be low awareness about the screening. For the intervention group, a low rate in the uptake of cervical cancer screening might be due to the short duration before the follow-up interviews where an average of six months was reported in other studies. The intervention would be stronger if it was assisted by more frequent sessions and reminders. Furthermore, our intervention targeted women who came to health facilities with their infants. This might discourage women to get screened especially right after they got the one to one health education. Most women had to go to their home to leave their children before getting screened.

One of the reasons for not getting screened for cervical cancer was the absence of disease symptoms. A considerable number of participants reported that they don't know about cervical cancer screening and the availability of screening services. This indicates low awareness about cervical cancer screening is a barrier to the uptake of screening services. This signifies the need for interventions that raise the community's awareness of cervical cancer and screening services.

Women who had university-level education were two times were likely to get screened. This finding is in line with studies conducted in Korea and Zimbabwe [19, 20]. Education increases the risk of awareness of people towards health risks. In addition, educated women have a better self- efficacy, and access to health care services. These might explain the increase in odds of screening among the educated. The same argument justifies the association between being a government employee and an increased odd of screening, as most positions in government offices require proper training. In contrast, studies done in developed countries indicated no association between educational status and cervical cancer screening [21]. This could be because of the fact that literacy level is high in those countries, unlike developing countries where nearly two-thirds of the participants in the intervention and control groups completed only primary and secondary education. In addition, more than half of the women in both groups were housewives. Hence, providing education about cervical cancer screening has a real difference in the uptake especially in the community with low educational levels.

This study also found that there is a significant association between history of sexually transmitted infections (STI) and cervical cancer screening uptake. Human papillomavirus (HPV), a sexually transmitted infection, is one of the strong predictors of cervical cancer. Health care service providers may emphasize on the need to screen for cervical cancer when women present with STIs. On the contrary, being tested for HIV was not associated with cervical cancer screening. HIV screening is routine testing to all pregnant women, nearly all women in the intervention and control groups were tested for HIV, this could mask the difference in screening for cervical cancer.

This study was not without limitation, the fact that we have measured the behavior (screening) after two months may underestimate the findings. On the other hand, this study was conducted in an urban area where most people can read and write or have someone around to read for them and could not be generalized to most parts of the country where about half of all women and a third of all men aged 15–49 were illiterate [22].

## Conclusion

This study suggested that, provision of printed media and brief and focused health education by a health professional at primary health facilities could increase the uptake of cervical cancer screening service in Addis Ababa health centers. We recommend that integrating a one-to-one health education and administering educational brochures which are written in simple languages about cervical cancer and benefits of early screening with the existing maternal health programs are worthwhile.

## Supporting information

**S1 Fig. Flow diagram of the progress through the phases of a parallel randomized trial of two groups.**
(TIF)

**S1 File.**
(DOC)

**S2 File.**
(DOCX)

## Acknowledgments

We are grateful for Women's Health Research Working Group at Addis Ababa University, College of Health Sciences for initiating this research. The Women's Health Research Working Group Members at the College of Health Sciences who commented the proposal at the inception of the project are well acknowledged.

## Author Contributions

**Conceptualization:** Selamawit Hirpa Abu, Berhan Tassew Woldehanna, Mahlet Yigeremu Gebremariam.

**Formal analysis:** Selamawit Hirpa Abu, Berhan Tassew Woldehanna, Etsehiwot Tilahun Nida.

**Methodology:** Selamawit Hirpa Abu, Berhan Tassew Woldehanna, Mahlet Yigeremu Gebremariam, Mitike Molla Sisay.

**Project administration:** Selamawit Hirpa Abu, Etsehiwot Tilahun Nida, Mitike Molla Sisay.

**Supervision:** Selamawit Hirpa Abu, Etsehiwot Tilahun Nida, Abigiya Wondimagegnehu Tilahun.

**Writing – original draft:** Selamawit Hirpa Abu, Abigiya Wondimagegnehu Tilahun.

**Writing – review & editing:** Mahlet Yigeremu Gebremariam, Mitike Molla Sisay.

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
