## [Decision Letter · Decision Letter 0]

9 Mar 2020

PONE-D-19-25303

Role of Health Education on Cervical Cancer Screening Uptake at Health Centers of Addis Ababa

PLOS ONE

Dear Mrs. Abu,

Thank you for submitting your manuscript to PLOS ONE. After careful consideration, we feel that it has merit but does not fully meet PLOS ONE’s publication criteria as it currently stands. Therefore, we invite you to submit a revised version of the manuscript that addresses the points raised during the review process.

We would appreciate receiving your revised manuscript by Apr 23 2020 11:59PM. To enhance the reproducibility of your results, we recommend that if applicable you deposit your laboratory protocols in protocols.io, where a protocol can be assigned its own identifier (DOI) such that it can be cited independently in the future. For instructions see: http://journals.plos.org/plosone/s/submission-guidelines#loc-laboratory-protocols

We look forward to receiving your revised manuscript.

Kind regards,

Stanley J. Robboy, MD

Academic Editor

PLOS ONE

Journal Requirements:

2. Please provide additional details regarding participant consent. In the ethics statement in the Methods and online submission information, please ensure that you have specified what type of consent you obtained (for instance, written or verbal, and if verbal, how it was documented and witnessed). If your study included minors, state whether you obtained consent from parents or guardians.

4. In your Methods section, please provide additional information about the participant recruitment method and the demographic details of your participants. Please ensure you have provided sufficient details to replicate the analyses such as: a) the recruitment date range (month and year), b) a description of any inclusion/exclusion criteria that were applied to participant recruitment, c) a description of how participants were recruited, and d) descriptions of where participants were recruited and where the research took place.

5. Please provide a sample size and power calculation in the Methods, or discuss the reasons for not performing one before study initiation.

6. Please amend the manuscript submission data (via Edit Submission) to include author Abigiya Wondimagegnehu and Mitike Molla.

7. Please amend your authorship list in your manuscript file to include author Mitike Sisay and Abigiya Tilahun.

8. Please amend either the title on the online submission form (via Edit Submission) or the title in the manuscript so that they are identical.

9. Thank you for stating the following in the Acknowledgments Section of your manuscript:

"We are grateful for Women’s Health Research Working Group at Addis Ababa University, College of Health Sciences for initiating and giving support for this research project. Also, we very thankful for ‘Impact of Maternal Death on Living children’ research project at Addis Ababa University, School of Public Health for funding this research."

10. Your ethics statement must appear in the Methods section of your manuscript. If your ethics statement is written in any section besides the Methods, please move it to the Methods section and delete it from any other section. Please also ensure that your ethics statement is included in your manuscript, as the ethics section of your online submission will not be published alongside your manuscript.

11. We note you have included a table to which you do not refer in the text of your manuscript. Please ensure that you refer to Table 3 in your text; if accepted, production will need this reference to link the reader to the Table.

Reviewers' comments:

Reviewer's Responses to Questions

**Comments to the Author**

1. Is the manuscript technically sound, and do the data support the conclusions?

Reviewer #1: Yes

Reviewer #2: Yes

2. Has the statistical analysis been performed appropriately and rigorously? 

Reviewer #1: No

Reviewer #2: Yes

3. Have the authors made all data underlying the findings in their manuscript fully available?

Reviewer #1: Yes

Reviewer #2: Yes

4. Is the manuscript presented in an intelligible fashion and written in standard English?

Reviewer #1: No

Reviewer #2: No

5. Review Comments to the Author

Reviewer #1: This study assessed the effect of providing cervical cancer prevention information on subsequent cervical cancer screening. The study is interesting particularly because of the dramatic increase in cervical cancer screening after a relatively simple intervention such as providing health education and giving brochures to women seeking routine care.

Major issues:

The article is poorly written. I suggest that authors use a professional language proofreading services if English is not their first language.

In the introduction section, the authors do not mention the current gold standard for cervical cancer screening, which is once every three years for women age 21 to 65 years, and once every five years for women age 30 to 65, if the screening is combined with HPV testing. The authors do not mention if the same screening guidelines are applicable in Ethiopia.

Methods: The authors do not explain why the study was limited to women aged 30 to 49 years.

The figure on the participant flow is confusing. It reads as if patients were allocated to the intervention and control health facilities. However, in the methods section, it states that facilities were selected randomly, and patients who visited those facilities received either the intervention or control based on the the facility they visited.

One major problem as well is the description of the control group. The authors do not describe what 'regular health education' means. They mention that education about CC is given twice a week. It is unclear what CC means.

Results:

The authors have provided descriptive information about the women who received the intervention and control. However, since the randomization was done at the facility level, I suggest that the authors conduct statistical tests to see if the treatment and control groups were statistically significant.

Minor issues:

Background: the authors should provide more recent statistics on the prevalence of cervical cancer globally and in Ethiopia, if available. Using figures from 2012 appears rather dated.

Reviewer #2: Overall, this is a well-designed study on a very important topic. My primary concern is that much of the paper is not grammatically correct. Editing for language and ease of reading would be helpful. The authors do a good job describing how their intervention affected uptake of cervical cancer screening and the population characteristics that affected whether subjects were screened. It would be helpful to have a more in-depth discussion of the identified barriers to screening an possible ways to overcome these barriers.

6. PLOS authors have the option to publish the peer review history of their article (what does this mean?). If published, this will include your full peer review and any attached files.

Reviewer #1: No

Reviewer #2: No

---

## [Author Response · Author response to Decision Letter 0]

3 Jun 2020

Response to academic editor and Reviewers 

Dear Reviewers, 

Thank you for your valuable comments. We have responded for each comment one by one 

Comment 

1- Please ensure that your manuscript meets PLOS ONE's style requirements

Response- We have ensured that the manuscript meets PLOS ONE’s style requirement 

2- Please provide additional details regarding participant consent. In the ethics statement in the Methods and online submission information, please ensure that you 

have specified what type of consent you obtained

Response- Now additional information is given (page 6)

3- Please include additional information regarding the survey or questionnaire used in the study and ensure that you have provided sufficient details that others could replicate the analyses. 

Response- The survey questionnaire developed for this study is now included as supporting information. 

4. In your Methods section, please provide additional information about the participant recruitment method and the demographic details of your participants. Please ensure you have provided sufficient details to replicate the analyses such as: a) the recruitment date range (month and year), b) a description of any inclusion/exclusion criteria that were applied to participant recruitment, c) a description of how participants were recruited, and d) descriptions of where participants were recruited and where the research took place.

Response- Comment accommodated (See methods section, page 4 and 5) 

5. Please provide a sample size and power calculation in the Methods, or discuss the reasons for not performing one before study initiation.

Response- Sample size calculation is now provided at the methods section (Refer page 6)

6. Please amend the manuscript submission data (via Edit Submission) to include author Abigiya Wondimagegnehu and Mitike Molla.

Response – Comment accommodated 

7. Please amend your authorship list in your manuscript file to include author Mitike Sisay and Abigiya Tilahun.

Response – Author Mitike Molla and Abigiya Wondemagne are already in the authorship list 

8. Please amend either the title on the online submission form (via Edit Submission) or the title in the manuscript so that they are identical.

Response- The title online submission and the title in the manuscript are identical 

9. Thank you for stating the following in the Acknowledgments Section of your manuscript:

"We are grateful for Women’s Health Research Working Group at Addis Ababa University, College of Health Sciences for initiating and giving support for this research project. Also, we very thankful for ‘Impact of Maternal Death on Living children’ research project at Addis Ababa University, School of Public Health for funding this research."

Response- Now, statement related to the funding is removed from the acknowledgement section. We have only put this in the funding statement 

10. Your ethics statement must appear in the Methods section of your manuscript. If your ethics statement is written in any section besides the Methods, please move it to the Methods section and delete it from any other section. Please also ensure that your ethics statement is included in your manuscript, as the ethics section of your online submission will not be published alongside your manuscript

Response- Ethics statement is already included in the methods section of the manuscript. 

11. We note you have included a table to which you do not refer in the text of your manuscript. Please ensure that you refer to Table 3 in your text; if accepted, production will need this reference to link the reader to the Table.

Response- Table 3 is now referred in the text 

12. Methods: The authors do not explain why the study was limited to women aged 30 to 49 years.

Response- Now it is explained in the methods section why 30-49 age group is only included. The reason is the Ethiopian Cervical Cancer screening Guideline recommends VIA screening for women in the age group of 30-49 years. 

13. The figure on the participant flow is confusing. It reads as if patients were allocated to the intervention and control health facilities. However, in the methods section, it states that facilities were selected randomly, and patients who visited those facilities received either the intervention or control based on the the facility they visited.

Response- From 14 primary health centres that were giving cervical cancer screening using VIA at the time the study period; only 8 health centres were randomly selected. From the eight health centres that were selected for this study; randomly 4 health centres in each arm were assigned randomly 

14. One major problem as well is the description of the control group. The authors do not describe what 'regular health education' means. They mention that education about CC is given twice a week. It is unclear what CC means.

Response -Now, CC is corrected to cervical cancer. 

Regular health education – is a health education done in all health facilities focusing on any health concern; mostly the education is done for a mass (People who come to the health facilities to get different health services). As part of this, cervical cancer is one of the health topics addressed at the health facilities in Ethiopia. 

15. The authors have provided descriptive information about the women who received the intervention and control. However, since the randomization was done at the facility level, I suggest that the authors conduct statistical tests to see if the treatment and control groups were statistically significant.

Response- we have conducted Chi- square test result of cervical cancer screening practice with type of health center (Intervention/Control) (As shown the in the table below)

Variable Screened for cervical cancer x2 value P- val-ue 

 Yes 

(n=211) No

(n=1926) 

Type of the health center Intervention 152 (70.70%)

 910 (47.27%) 

 42.4519 

 0.000

 Control 63 (29.30%) 1,015 (52.73%) 

Proportion of women screened for cervical cancer among the intervention group was higher than the control group. And there is association between receiving health education and educa-tional brochure with cervical cancer screening practice.

16. The authors should provide more recent statistics on the prevalence of cervical cancer globally and in Ethiopia, if available. Using figures from 2012 appears rather dated.

Response- Comment accommodated. Now 2018 data is given about the prevalence of cervical cancer screen-ing ( See Introduction section) 

17. My primary concern is that much of the paper is not grammatically correct. Editing for lan-guage and ease of reading would be helpful. 

Response- We have done grammatical correction and edited for language ease for reading. 

18. The authors do a good job describing how their intervention affected uptake of cervical cancer screening and the population characteristics that affected whether subjects were screened. It would be helpful to have a more in-depth discussion of the identified barriers to screening and possible ways to overcome these barriers.

Response- Comment accommodated (Page 15, paragraph 2)

---

## [Decision Letter · Decision Letter 1]

2 Jul 2020

PONE-D-19-25303R1

The Role of Health Education on Cervical Cancer Screening Uptake at Selected Health Centers in Addis Ababa

PLOS ONE

Dear Dr. Abu,

Thank you for submitting your manuscript to PLOS ONE. After careful consideration, we feel that it has merit but does not fully meet PLOS ONE’s publication criteria as it currently stands. Therefore, we invite you to submit a revised version of the manuscript that addresses the points raised during the review process.

The topic of cervical cancer screening is important and the authors are encouraged to address the concerns raised in the reviews.  Given that the current submission is itself also a revision, I have served this time as one of the new reviewers.  When resubmitting, please have someone thoroughly familiar with English proofread the manuscript for language consistancy.  Plos One itself provides no editing, but can provide names where editing assistance is available.

We look forward to receiving your revised manuscript.

Kind regards,

Stanley J. Robboy, MD

Academic Editor

PLOS ONE

Reviewers' comments:

Reviewer's Responses to Questions

**Comments to the Author**

1. If the authors have adequately addressed your comments raised in a previous round of review and you feel that this manuscript is now acceptable for publication, you may indicate that here to bypass the “Comments to the Author” section, enter your conflict of interest statement in the “Confidential to Editor” section, and submit your "Accept" recommendation.

Reviewer #3: (No Response)

Reviewer #4: (No Response)

2. Is the manuscript technically sound, and do the data support the conclusions?

Reviewer #3: Partly

Reviewer #4: Partly

3. Has the statistical analysis been performed appropriately and rigorously? 

Reviewer #3: Yes

Reviewer #4: I Don't Know

4. Have the authors made all data underlying the findings in their manuscript fully available?

Reviewer #3: Yes

Reviewer #4: Yes

5. Is the manuscript presented in an intelligible fashion and written in standard English?

Reviewer #3: No

Reviewer #4: No

6. Review Comments to the Author

Reviewer #3: This study examined the effect of health education on cervical cancer screening uptake in 2,139 women at selected health facilities in Addis Ababa, Ethiopia, where awareness of the disease and availability of screening and treatment are limited. A RCT was conducted at eight public health care centers that provide cervical cancer screening services using visual inspection with acetic acid; each center was assigned to either intervention (one-on-one health education, and receipt of a brochure explaining cervical cancer and benefits of screening) or control (regular care). After two months, women were contacted to ask whether they had been screened for cervical cancer.

This paper deals with a very important topic, and an easily and inexpensively implemented intervention, but given the results, it is NOT effective. The authors found a significant effect in the intervention group; however, they need to address more fully the obvious problem that even in the intervention group, there was only 14% uptake of services. The intervention may work better than nothing, but it isn’t working very well if >85% of the women who received the intervention did not seek screening. Although the authors surmise what the reason for the low rate of uptake might be, they do not take the opportunity to put forward ideas about what might have been done differently or what might have been wrong with the intervention.

Was any attempt made to schedule an appointment for the women at the time they were enrolled, or at any time after the intervention? Education helped but was in no way enough to motivate these women; it either missed the mark in the way the information was conveyed, or did not include enough information to be taken seriously. Scheduling an appointment on the spot would be far more effective than leaving it up to these women themselves, who are older and therefore probably assume they have managed to go without screening this long without any bad effects, so “why bother?”

At two months, the majority of the women in the intervention group (almost 80%) said that the primary reason for not seeking screening was because they “were busy;” in person, this sort of excuse could be directly addressed. The majority of the women are housewives, with presumably older children (given their age range). If the intervention included an attempt to schedule an appointment, perhaps this “too busy” issue could be directly addressed with these women and solutions found (were they really “too busy,” or did they just need further nudging, reassurance and a stronger emphasis on the importance of screening?). If this study is to be valuable to future researchers, the authors should put more emphasis on what went wrong, why, and how it might be fixed.

The study needs to be edited by a native English speaker; there are many incomplete or incoherent sentences and grammatical errors.

Other comments:

Abstract:

[Methods]

“The eight health centers were stratified into two equal numbers of groups as intervention and control.” A more clear way to state this would be: Each of the eight health centers was randomly assigned to serve as either an intervention or a control center.

Introduction

Delete the reference to global cases of cervical cancer; it isn’t really relevant. Should just read, “In 2018 alone, 6,294 new cases of cervical cancer occurred in Ethiopia…”

Methods

A table for the sample size calculation really isn’t necessary. This information can be included in the text.

[Data entry and analysis] First sentence should read “Data entry and data analysis were done using….respectively.”

Table 1.

The authors should consider testing the differences between the two cohorts, given that the groups each come from four different centers.

Table 2.

Could the authors address whether they had difficulty contacting the women after two months? It appears from Table 2 that follow-up was very good but not complete. What efforts were made to contact the women who could not be reached? The authors should summarize this information in the results (including reporting the percentage of incomplete assessment in each group). It would be interesting to compare statistically the reasons that women gave for not being screened between the two groups (simple chi-square tests could be used).

[Multivariable analysis]

The way in which the authors describe the modeling process needs to be improved. Their main predictor is group status; the other variables in the model should be described as variables for which they adjusted the primary predictor in the model, based on the bivariate results. The emphasis should be on the intervention/control group status. This is buried in the paragraph, which is not a very good description of the steps that went into the analysis. This section could be more detailed.

Table 3.

The title of the table should not be “Multivariable Analysis…” if it also contains the bivariate analysis (I am assuming this is what the “Crude OR” refers to). The unadjusted analyses should show results of all models, even those for covariates that did not turn out to be significantly associated with the outcome. The columns for variables not included in the multivariable analysis would just be left blank in the “Adjusted OR” columns. Group status should be listed first. P-values should be reported for both the unadjusted and adjusted models. There is a “1” missing (as reference) for Marital Status for Crude OR.

Discussion

7% of the intervention group reported that they were not screened because they didn’t know about cervical cancer screening and availability of the services. This is surprising in the group that received the intervention and indicates some failing of the intervention for them. Did the authors pursue this, to determine why these women claimed to lack awareness?

This is an important paper which should be published, but it still needs some work.

Reviewer #4: PONE-D-19-25303R1

Reviewer comments for the authors.

ABSTRACT

Several phrases in the abstract or in the underlying main article are unclear and are noted below

Methods

one-to-one brief health education : Need further description. The One to one brief health education in the interventional center is obvious. How does health education in the interventional center differ from the non-interventional (control) center? A clearer description is needed describing the differences of the client receives at the interventional vs non-interventional centers.

pectoral description : “Pectoral” refers to the chest muscles; I believe the authors mean “pictures.”

After two months : Did clients sometimes have the treatment at the time of the initial visit or did they actually have to go home and then return for a special cancer screening

called through their phones : Awkward. Called by phone somehow sounds better

Results

215 (10%) : Medical care in Addis Ababa. The intervention essentially doubled the number of people screened for cervical cancer, yet the overall rates remained low (10% of all women in the study). The goal, ideally, is for a 100% rate of screening. Am I correct only about 3% (64) of the non-interventional cluster came for screening? The discussion section of the paper might wish to offer thoughts as to why the rates for both groups remained so low.

first degree or above : To what does first degree refer: education?

STI: Spell out

Introduction

3rd Paragraph: The expansion …

study done among 1186 women in Mekelle : Interesting that the rate of screening at Mekelle was roughly 6 times greater there than at your institution (19.6% vs 3% of your noninterventional cluster). What is the difference in your two populations

5th para: In our study …

theoretical : I do not believe you mean “theoretical,” more likely the word should be “practical”

Methods

Study area and period

The study was conducted in Addis Ababa, the capital city of Ethiopia between August 2017 and January 2018.: the sentence is constructed awkwardly. The authors mean the study was conducted between August 2017 and January 2018, but as written states that Addis Abba was the capital between those two dates.

tire: Tier, not tire

Unlike other regions in the country,: Why are the Addis Ababa health centers unlike those of other parts of the country? Do the others lack general practitioners, nurses, or midwives? Also, for those of us unfamiliar with the health system in Ethiopia, what are the key characteristics of the second level and tertiary level care providers?

Study design and population

Two-pronged a cluster: Do you mean “a two-pronged” rather than “two-pronged a”?

30-49 years. Which is : I believe you mean “30-49 years, which is” rather than a new sentence “30-49 years. Which is”

under 5 clinics for their babies : I think you mean “babies and children under five years of age”

Women who gave birth in the past 45 days and pregnant : awkward wording. The sentence reads as if someone had recently given birth but is still pregnant.

Intervention group:

one-to-one brief health talk : Given one to one talk. How was the brochure the interventional group received different from that the control clients received? Within the interventional group, what was more effective, the personal talk, or whatever reading material the client received?

Each woman who participated in the intervention group : Would it be correct to assume that women in the intervention group did NOT have the screening performed at the same time as they came for the initial interview? Did many of the women at the initial interview schedule their return visit for VIA or did they return home, and once there, schedule a visit? This seems to me to be important in determining what type of a future educational campaign might be waged. If the former, more women need to be encouraged to come to the centers and have one to one interviews. If the latter, then the campaign may consist more of brochures sent out throughout the community

Control group:

except the regular health education : It is unclear of what “regular health education” consists.

What exactly is the difference between “regular” in the controls and the brochures in the interventional group. This would seem to be an important facet to clarify in a study of this type. Some more detail would be helpful.

Ethical consideration:

Each respondent was informed about the purpose, scope, and expected the outcome of the research and verbal informed consent as it was approved : Unclear: Were control told they would not be given 1-1 counseling?

“Expected the outcome” -- I think you mean “Expectations of the outcomes”?

The data collectors put a mark on the checklist given on the consent form, regarding the study participants' responses to participate. : Are the authors’ saying checkmarks were put on all clients forms, so that at the end of the study the authors could determine the total number asked, and thus also what percent of those asked to join the study, regardless whether intervention or control, actually refused? I presume that is from where the number, 2139, derives.

The study is also hazy about what “participates” means, which affects the study numbers, and hence percentages reported.

When a woman came to Bole-17 for example, was she asked if she would participate, and if so, only then did she receive a one-on-one session?

Also, was VIS performed on the same day and possibly even at the same time as the interview, or did clients have to return on a subsequent day, which certainly would reduce compliance and therefor the number of study participants completing all desired aspect?

Generalized estimating equation (GEE) analysis with a binary response variable using a robust estimator and exchangeable working correlation matrix was used.: I am not an epidemiologist or statistician and have no idea what the sentence means. Can this be simplified and expressed correctly, but with words I might understand?

Result

A total of 2,139 women who had never screened : Add the missing word “been” after “never”

During the study period,: Until this point, the two populations seem relatively similar. The rest of the paragraph left me stunned, and unprepared to understand how the population as a whole differed from both the interventional and control clients. (Actually the answer came only late in the discussion – ALL pregnant women are routinely tested for HIV. Thus, nearly 100% of the clients have been tested for HIV. Is this normal for Africa, for Ethiopia, specifically for Addis Ababa, or specifically for the population of patients with which your group deals? Somewhere earlier, there needs to be a description, however brief, of your population.

The remainder of the paragraph should present the data to help the reader. Possibly the authors might wish to present the HIV data, then the STI data and finally the cervical data.

In the discussion, somewhere, you may wish a few comments about how your group is forward-looking and trying to advance medical care in Ethiopia in ways that are economically practical.

Regarding the health status of participants,: My comments refer to this paragraph and also table 1. The single most striking fact of table 1 is that over 90% of all your clients, regardless of whether intervention or control, were tested for HIV, now or in the past. From the discussion, I presume this is routine.

With this high rate of HIV testing, why is there not greater concern about STI's and cervical cancer? It seems cervical cancer is relatively ignored. Is this because the women generally know nothing about cervical cancer and do not understand it? Is this part of the overall education programs your medical centers/leaders are trying to bring to the Ethiopian citizenry? The discussion might expand on this subject.

STI. Where 1,005(95%): Spell out STI as it is the first time of use in the text.

I believe the two phrases belong to one sentence.

Table 1. Socio demographic and Health Related Information of Study Participants at Addis Ababa, 2017

Diploma

First Degree and above: What is the difference between diploma and first-degree; Will most readers understand?

Religion

Orthodox : Are not the orthodox part of the Christian community? Would it be clearer, given a worldwide readership likely unfamiliar with Ethiopian religion to refer to the Orthodox as Ethiopian Christian Orthodox? This would avoid confusion with Jewish Orthodox.

All study participants … telephone about the source of information after two months of time from the recruitment period: This sentence is problematic. The study has two arms where patients received specific one-on-one counseling and brochures, or little other than the general information provided by the clinic. But now a third arm is suggested, where the patients receive new sources of information during the subsequent telephone calls. Does this mean some clients in the interventional or control groups learn from a third source and were acting on that new information? Do we now have a murky third arm? Please explain.

Discussion

This study was conducted in an urban area where most people can read and write or have someone around to read for them : Useful in methods to help describe the population clientele.

Paragraph:

In this study, women who were screened for cervical cancer reported that the health education they have received from one-to-one health education and the educational brochure had helped them to decide in screening for cervical cancer. : Was there anything the healthcare professionals learned when giving advice to getting the clients attention and having the client follow through with testing? If so, please add in the results section and comment in the discussion section.

Paragraph: This study also found a significant association between history of STI

. This could be related to the fact that HIV screening is a routine procedure that is provided for all pregnant women at all health facilities hence: I was unaware of this fact, which explains my comment related to table 1. Please comment about routine HIV screening in pregnant women much earlier.

References

: Please refer the Journal’s instructions; Most references are incomplete

#2 : Who is the author, what is the journal or book etc.

#3 : Is this a web reference. Need URL & Date access

#4 : Web site note proper – too many periods.

#5 : Citation incomplete

Etc.

7. PLOS authors have the option to publish the peer review history of their article (what does this mean?). If published, this will include your full peer review and any attached files.

Reviewer #3: No

Reviewer #4: **Yes: **Stanley J Robboy, MD

---

## [Author Response · Author response to Decision Letter 1]

15 Aug 2020

We have addressed the commentes given

---

## [Editor Report · Decision Letter 2]

10 Sep 2020

The Role of Health Education on Cervical Cancer Screening Uptake at Selected Health Centers in Addis Ababa

PONE-D-19-25303R2

Dear Dr. Abu,

We’re pleased to inform you that your manuscript has been judged scientifically suitable for publication and will be formally accepted for publication once it meets all outstanding technical requirements.

Kind regards,

Stanley J. Robboy, MD

Academic Editor

PLOS ONE

Additional Editor Comments (optional):

The revisions submitted are good.

Attached is a file the Academic Editor has made as suggested changes to the manuscript.

As the Journal does not make changes to submitted manuscripts, I have made some suggested wording changes that the authors may wish to include.

---

## [Editor Report · Acceptance letter]

22 Sep 2020

PONE-D-19-25303R2 

The Role of Health Education on Cervical Cancer Screening Uptake at Selected Health Centers in Addis Ababa 

Dear Dr. Abu:

I'm pleased to inform you that your manuscript has been deemed suitable for publication in PLOS ONE. Congratulations! Your manuscript is now with our production department. 

Kind regards, 

on behalf of

Dr. Stanley J. Robboy 

Academic Editor

PLOS ONE